# Influence of Humidity on NO₂-Sensing and Selectivity of Spray-CVD Grown ZnO Thin Film above 400 °C

**Roussin Lontio Fomekong and Bilge Saruhan \*** 

German Aerospace Center, Institute of Materials Research, Department of High-Temperature and Functional Coatings, 51147 Cologne, Germany; Roussin.LontioFomekong@dlr.de

\* Correspondence: bilge.saruhan@dlr.de

**Abstract:** Thin films are being used more and more in gas sensing applications, relying on their high surface area to volume ratio. In this study, ZnO thin film was produced through a thermal aerosol spraying and chemical vapor deposition (spray-CVD) process at 500 °C using zinc acetate as a precursor. The phase identification and the morphologies of the film were investigated by XRD and SEM, respectively. Gas-sensing properties of the ZnO thin film were evaluated toward NO₂, CO, and NO at a moderate temperature range (400–500 °C) in dry and humid air (relative humidity = 2.5, 5, 7.5, and 10% RH). The obtained results show good sensor signal for both NO₂ ($\Delta R/R_0$ = 94%) and CO (92%) and poor sensor signal to NO (52%) at an optimum temperature of 450 °C in dry air. The response and recovery times decrease with the increase of NO₂ concentration. In the presence of humidity (10% of RH), the sensor is more than twice as sensitive to NO₂ (70%) as CO (29%), and accordingly, exhibits good selectivity toward NO₂. As the amount of humidity increases from 2.5 to 10% RH, the selectivity ratio of ZnO thin film to NO₂ against CO increases from 1 to 2.4. It was also observed that the response and the recovery rates decrease with the increase of relative humidity. The significant enhancement of the selectivity of ZnO thin film toward NO₂ in the presence of humidity was attributed to the strong affinity of OH species with NO₂.

**Keywords:** ZnO; thin film; spray-CVD; NO₂ sensor

## 1. Introduction

In recent years, the air pollution-related problems caused by combustion systems at factories and exhaust gases from automotive vehicles, etc., became serious and threatening for the world population. In particular, the focus lies on NOx because a high NOx content in the atmosphere has various consequences such as acid rain as well as reduction of the ozone layer. Despite the fact that strict environmental and exhaust control standards have been released and applied in many countries, the NOx content could not be drastically reduced. NO₂ is a toxic gas and the cause of inflammation of lung tissue, bronchiolitis fibrosa obliterans, and silo-filler's disease. NO₂ emitted by supersonic jets in atmosphere causes the destruction of ozone layer present in the stratosphere which absorbs the harmful damaging UV radiation coming from the sun [1]. Therefore, inexpensive, small-sized, and maintenance-free NOx sensors for air-quality monitoring (operating at low or room temperature), emission-control systems, and automotive exhausts application (operating at temperatures above 400 °C) are needed.

The most common and practical sensors are the metal oxide semiconductor sensors that have many advantages, such as good working stability, high sensitivity, simple manufacturing process, etc. Common metal oxide semiconductors include ZnO, SnO₂, TiO₂, CuO, Fe₂O₃, WO₃,

Ga$_2$O$_3$, and V$_2$O$_5$ [2–7]. Among those, ZnO belongs to the II–VI group compound exhibiting the characteristics of n-type semiconductor with a wide direct band gap of 3·3 eV at room temperature [8]. ZnO has recently gained attention as a gas sensing material due to its high conductive electron mobility and excellent chemical and thermal stability under sensor operating conditions [6]. In addition, ZnO possesses various excellent properties such as high electrochemical stability, low cost, nontoxicity, etc. [9]. Due to its advantages of high sensitivity, stability, and low cost, ZnO nanostructures have been widely used for detecting gases at low or room temperatures such as O$_3$, NH$_3$, NO$_2$, and ethanol [10–13]. However, its operating sensing temperature is generally limited to 400 °C [1,14]. Beyond 400 °C, a drop of sensitivity is usually observed. The poor selectivity restricts also the applicability of ZnO based gas sensors. So far, some strategies including surface functionalization with noble metals, designing hetero-structures, fabrication of nanocomposites, utilization of zeolite, thermal assistance with UV-illumination, etc. have been developed to improve the sensor selectivity for a given gas species [15–17]. For exhaust gas emission-control systems for automotive applications, good sensitivity and selectivity are required at high gas temperatures (>400 °C).

Nowadays, metal oxide thin films are receiving tremendous attention for various microelectronics applications such as optoelectronics, ultraviolet (UV) light emitters, piezoelectric devices, gas sensing, etc. [18,19]. As a matter of fact, for sensing applications, in contrast to the bulk form, the thin film form with its high surface area to volume ratio is expected to be most sensitive since sensing is basically a surface phenomenon [20]. The synergic combination of the excellent sensing performance of ZnO with the potentiality of 2D nanostructures has resulted in high detection efficiencies and fast response and recovery capability. Various physical and chemical techniques have been utilized to obtain ZnO thin films [14]. The physical techniques include thermal oxidation, reactive evaporation, electron beam evaporation, different forms of sputtering (e.g., magnetron sputtering, rf sputtering), etc. On the other hand, CVD, sol-gel, and spray pyrolysis are mostly used chemical techniques to synthesize gas-sensitive layers of ZnO. Even though good sensing properties have been reported for ZnO thin film, only few works reported its sensing properties beyond 400 °C [1,21,22]. Moreover, the influence of humidity which is present in the mentioned application environments should be considered seriously for on sensing performance of ZnO thin film under real applications at high temperature. This aspect is not reported in the literature.

To the best of our knowledge, we report the influence of relative humidity on sensing performance (selectivity) of ZnO thin film beyond 400 °C in this work for the first time. ZnO film was deposited directly on interdigital electrodes (IDE) substrate by spray-CVD. The influence of humidity was also investigated on sensitivity, selectivity, and on response and recovery rates.

## 2. Materials and Methods

### 2.1. Deposition of ZnO Film on Sensor Platform

The spray-CVD (from Co. Annealsys, France) is a relatively cheap and simple technique where the precursor of the material is deposited from solution by spraying onto a heated substrate using air and nitrogen as carrier gas. The decomposition of zinc acetate occurs on the substrate following the reaction:

$$Zn(CH_3COO)_2 \text{ (g)} + 4O_2 \text{ (g)} \xrightarrow{450\ ^\circ C} ZnO \text{ (s)} + 4CO_2 \text{ (g)} + 3H_2O \text{ (g)}$$

The gaseous byproducts escape from the surface very easily.

The spray precursor used in this study was 0.1 M zinc acetate [Zn(CH$_3$COO)$_2$·2H$_2$O] (99.99% pure) solution prepared in methanol. A few drops of acetic acid were added to prevent the hydrolysis of the acetate solution. The applied spray rate was 5 mL/min. The substrate (e.g., a sensor platform) was made of a ceramic fitted with gold interdigital electrodes (IDE) and was used without specific cleaning. The substrate temperature was kept over the heater at 500 °C through the IR lamps. A deposition time of 15 min was selected.

## 2.2. Characterization

The X-ray powder diffraction investigations were carried out using a Bruker D8 GADDS X-ray diffractometer equipped with cross-coupled Göbel mirrors and a two-dimensional GAADS detector system utilizing CuKα radiation (λCuKα = 0.15418 nm). The reflections from the JCPDS database were assigned to the experimental diffractograms with the program EVA from BRUKER AXS.

The morphology of the particles was determined by scanning electron microscopy (SEM) analysis and was carried out in a Zeiss Ultra 55 microscope equipped with an energy-dispersive X-ray spectrometer (EDS) from Oxford Instruments.

## 2.3. Gas Sensing Test

ZnO was deposited as thin films using thermal spray-CVD method on alumina substrates fitted with interdigital electrodes (IDE). The Pt-interdigitated electrodes were provided by the Co. Siegert Thin Film Technology Gmbh in Hermsdorf/Germany by fitting through lift-off technique. The interdigital design consisted of 10 interdigital bars of 300 μm wide and 2 mm long that had a gap of 300 μm between each other. The final thickness of the Platine electrode coating was approximately 1 μm. The sensor measurements were carried out in a specially constructed apparatus consisted of an eight-channel mass flow controller (that is used to adjust the gas concentration) from MKS Instruments GmbH (MFC-647b) followed by a gas mixing chamber consisting of a CARBOLITE tube furnace with a quartz-glass recipient and DC-measurement unit from a Keithley 2635A Sourcemeter. The sample was mounted on a sample holder and then placed in a quartz-glass recipient with no heater at the back of the sensor sample. Sensor tests are carried out in a quartz glass recipient heated by a tube furnace, the temperature of which is controlled over three cascades and adjusted by a thermocouple positioned in the quartz recipient at a distance of 2 cm from the sensor surface. The quartz-glass recipient has a 3 m long spiral through which the test gas is sent and heated by the same furnace in order to avoid cooler gas contacting the sensor surface. The whole system including the gas inlet and sensor chamber is kept heated during the sensor test. Prior to gas-sensing measurements, a warm-up heating was employed at the required testing temperature (i.e., 400–500 °C) for 1 h under airflow to achieve chemical stabilization of the sensor surface and a steady baseline resistance. All measurements were done in synthetic airflow at a rate of 400 mL/min with a constant current of $1 \times 10^{-6}$ A and voltage of 1 V. The moist air was obtained by humidification of dry air through a water bubbler. The sensor response for n-type semiconductors is defined by $S = (R_{gas}/R_{air} - 1) \times 100$ and $(R_{air}/R_{gas} - 1) \times 100$ for oxidizing and reducing gases, respectively.

## 3. Results and Discussion

The crystal structure and the purity of the as-deposited thin film were investigated by X-ray diffraction (XRD) and presented in Figure 1. As displayed, all the diffraction peaks, (100), (002), (101), (102), (110), (103), and (112), can be indexed as single-phase, hexagonal wurtzite structured ZnO according to JCPDS card No. 36-1451 [23]. No additional diffraction peaks were detected in the XRD patterns, indicating that spray-CVD deposited thin film yields pure phase. The presence of diffraction peaks along different planes indicates that the ZnO crystals have grown up in a multidirectional manner.

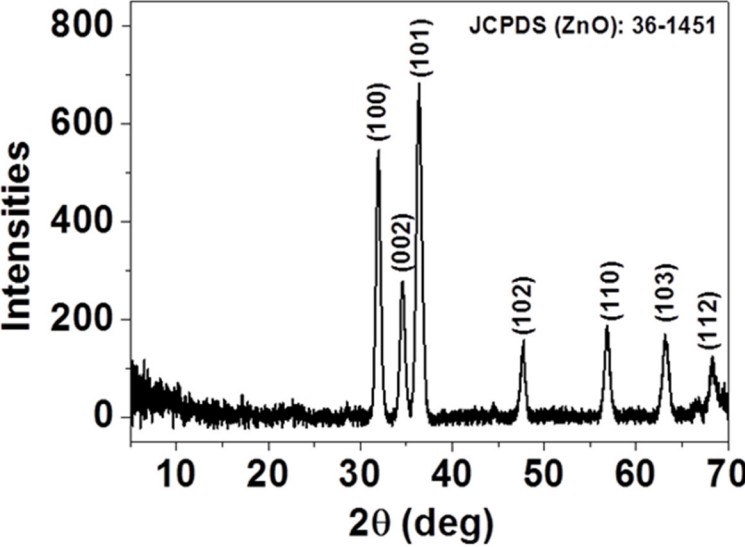

**Figure 1.** XRD of as-deposited ZnO thin film.

Analysis of microstructure, carried out by SEM, exhibits that the ZnO thin film consists of hexagonal-structured flake-like morphology with some agglomerated accumulation. Nevertheless, the film morphology yields a dense and quite homogenous appearance as shown in Figure 2. Our closer observation of the entire ZnO-coated substrate surface exhibited that the particles were densely stacked in the part of the substrate that was close to the spray nozzle, yielding a highly uniform morphology and thicker film. The mean value of the thickness all around the substrate was estimated to be around 2.5 μm by SEM depth measurements.

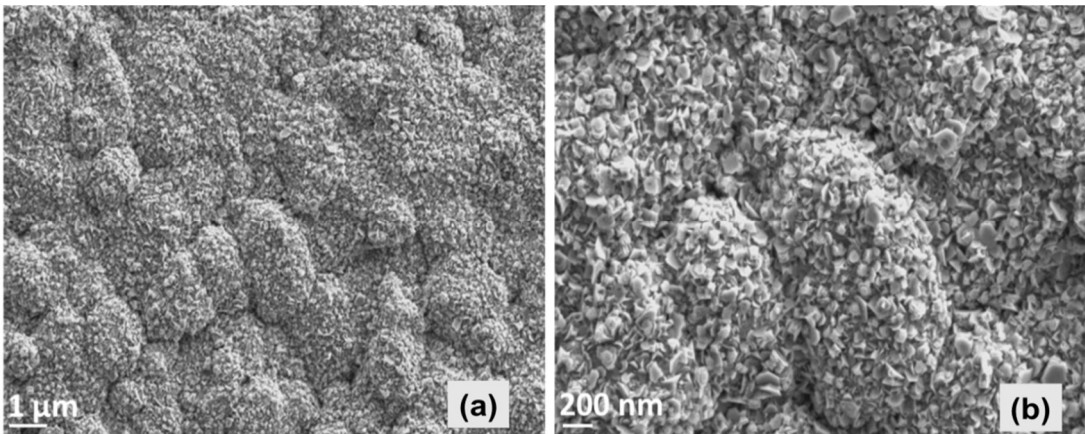

**Figure 2.** SEM images displaying the top view morphology of ZnO thin film at low (**a**) and high (**b**) magnifications.

The working temperature is an important parameter for gas sensor materials due to its significant influence on the occurrence of surface reactions during gas-sensing process. In order to determine the optimum operating temperature, the responses of the sensors prepared with ZnO thin films to 200 ppm $NO_2$ were measured in dry air at temperatures varying from 400 to 500 °C. The gas sensing results are shown in Figure 3 which displays that the response increases as the operating temperature increases up to 450 °C but then decreases with further increase of the operating temperature to 500 °C. This implies that the maximum response is reached with 94% at 450 °C, indicating that the optimum operating temperature of this ZnO sensor is 450 °C. This statement can be explained as follows. Firstly, it is well-known that the operating temperature is the source for achievement of the activation energy

required to complete the reaction for reduction of NO$_2$. With the increase of the operating temperature, more energy is provided for NO$_2$ reduction, resulting in an increase of the sensor signal with ZnO. The decrease of sensor response observed at operating temperature above 450 °C could be attributed to the phenomenon given by the Langmuir adsorption probability theory. On the increase of temperature above that is required for the optimum activation energy, the desorption rate of NO$_2$ on the ZnO film surface becomes higher than its adsorption rate, which leads to a decrease in gas sensitivity.

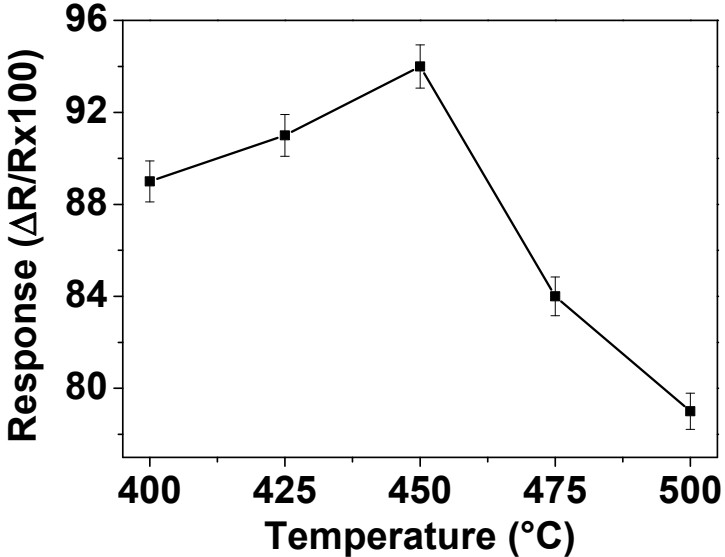

**Figure 3.** Response of gas sensor based on ZnO thin film to 200 ppm of NO$_2$ at different operating temperatures in dry air.

Figure 4 displays the dynamic response of the sensor with ZnO film at 450 °C to various NO$_2$ concentrations (e.g., 25, 50, 100, and 200 ppm) in dry air. As it can be seen, upon gas exposure, the sensor reaches an equilibrium resistance value and settles back to original base resistance value when the test gas is vented. This sequence has been reproduced for all the applied NO$_2$ concentrations, demonstrating the excellent reproducibility and stability of the sensor behavior. The sensor signal was calculated as 73, 86, 91, and 94% for 25, 50, 100, and 200 ppm of NO$_2$, respectively. These investigations indicate that the gas response increases with increasing concentration of NO$_2$ suggesting the capability of the sensor for quantitative analysis. In order to confirm reliability and reproducibility of the obtained ZnO film at the same NO$_2$ concentration, the sensor tests were repeated by introducing four times 50 ppm NO$_2$ into the test chamber under the same test conditions. The dynamic curve of these investigations is shown in Figure 4b. The average value of the gas response is 86%, with a small negligible deviation. The response/recovery times are quite similar. These results allow assuming that the NO$_2$ sensor prepared with Spray CVD produced ZnO thin film yields pretty good reproducible sensing results.

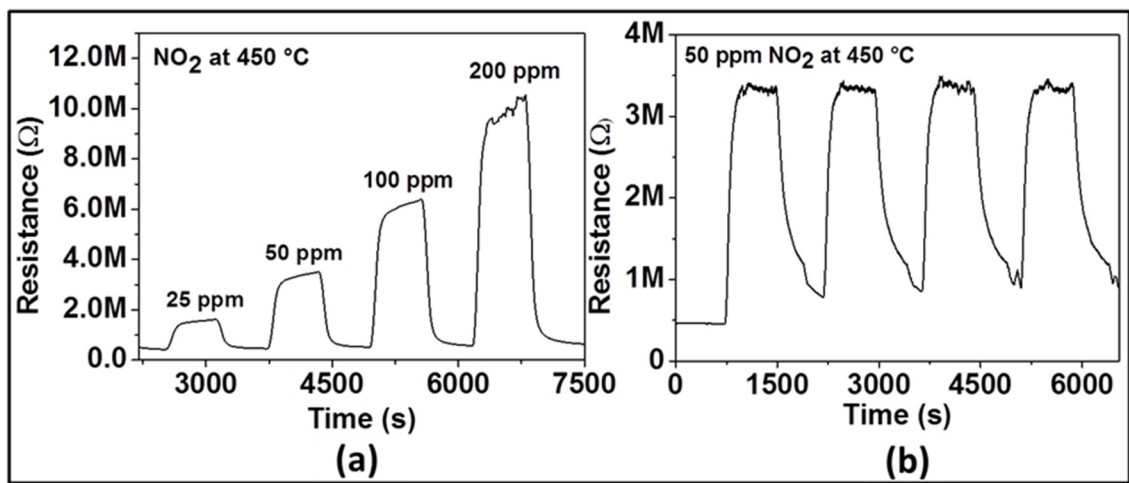

**Figure 4.** Dynamic response of gas sensor based on ZnO thin film to various concentrations of $NO_2$ (**a**) and to 50 ppm $NO_2$ (**b**) at an optimal operating temperature of 450 °C.

Response and recovery times are important sensor characteristics for real-time measurement. Therefore, we measured the response and recovery times of ZnO thin film in dry air for different $NO_2$ concentrations and defined these by taking the time that is required for the gas sensor to achieve 90% of the total resistance change in the cases of adsorption and desorption, respectively. As shown in Figure 5, the response time decreases from 205 to 162 s and the recovery time decreases also from 164 to 134 s when the $NO_2$ concentration increases from 25 to 200 ppm. A similar trend has been reported in the literature [24]. This can be due to the basic effects of surface covering kinetics and diffusion. At low $NO_2$ concentration, the covering rate of the surface is low (i.e., more time is needed to cover the entire surface), resulting in long response time. As the $NO_2$ concentration increases, the rate of surface coverage increases resulting in the decrease of response time.

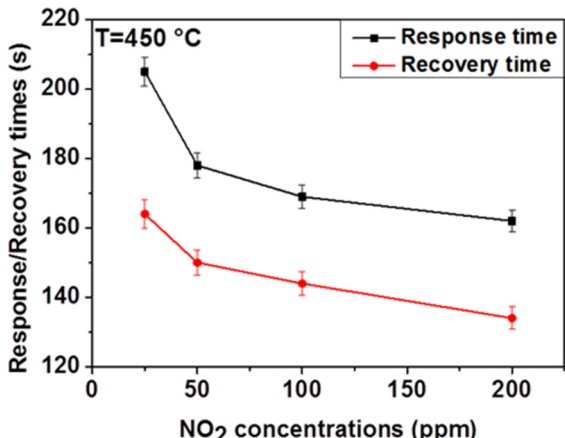

**Figure 5.** Relationship between $NO_2$ concentrations and the response and the recovery time for the ZnO thin film at 450 °C in dry air.

For practical applications, gas sensors are required not only to have good sensitivity and reproducibility, but also high selectivity to the target gas. Therefore, in order to evaluate its selectivity at the optimum operating temperature (450 °C), the response of the sensor with ZnO thin film was explored toward two other main exhaust gases (e.g., CO and NO). As observed in Figure 6a, the response of this sensor to 200 ppm of $NO_2$ (94%) is higher than that of 200 ppm of NO (52%) and almost the same as that of 200 ppm of CO (92%). This may imply for the sensor's selectivity to $NO_2$ in the presence of NO at 450 °C, however, this renders in the presence of CO. These results can be

explained by the variation of the activation energy with the interaction involving these applied test gases at the operating temperature. $NO_2$ and CO have probably activation energies closer to each other for their reduction and oxidation processes, respectively, while the activation energy differs in the case of the interaction with NO. The dynamic responses of the sensor to the three gases are shown in Figure 6b and confirm the n-type nature of ZnO. This is because the resistance increases (due to the reduction reaction) in the presence of $NO_2$ and NO, while the resistance decreases (due to the oxidation reaction) in the presence of CO.

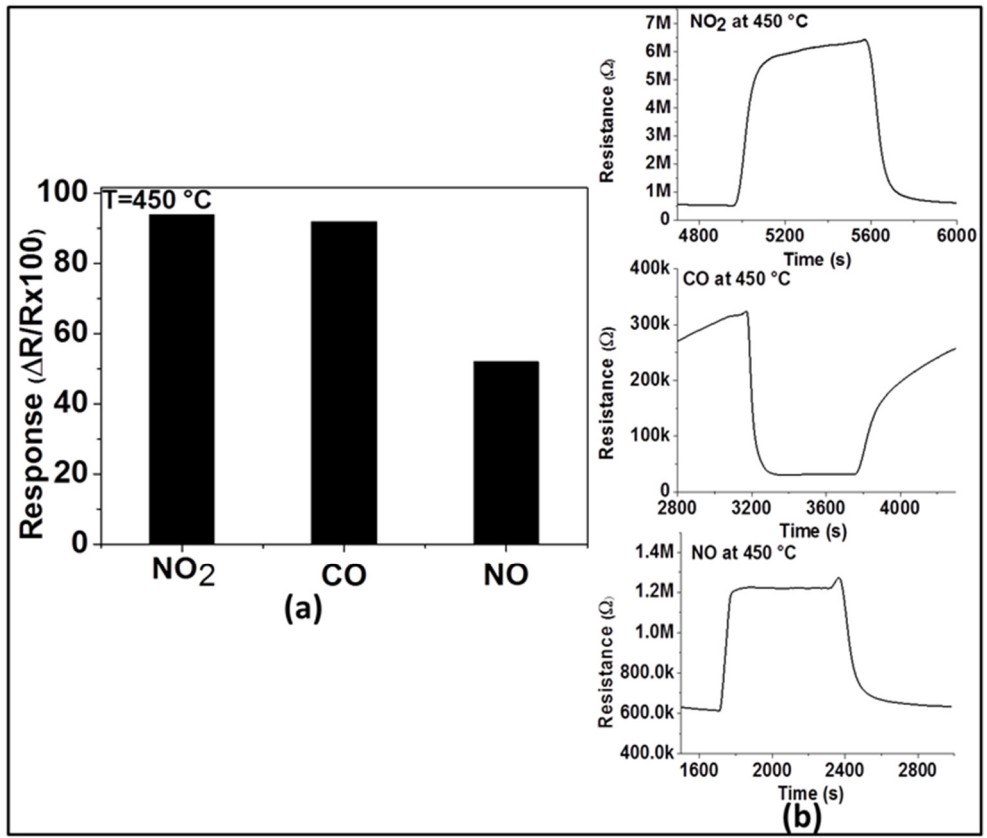

**Figure 6.** Response of gas sensor based on ZnO thin film to various gases including 200 ppm of $NO_2$, CO, and NO at 450 °C (**a**) and their dynamic responses (**b**).

Under dry conditions, the selectivity of the sensor with ZnO thin film to $NO_2$ was very poor if CO was also present. Usually, such sensors are being operated in humid environments, because of the presence of water vapor in the exhaust gas. Relying on our previous observations, we have investigated the effect of humidity on the sensing of $NO_2$ and CO with the ZnO sensor at 450 °C. The response of the sensor toward $NO_2$ and CO in the presence of different contents of relative humidity was investigated at 450 °C and the results are presented in Figure 7. As Figure 7a indicates, the $NO_2$ and CO sensor signals decrease with the increase of relative humidity. As far as $NO_2$ is concerned, the measured sensor responses were 94, 81, 76, 73, and 70% for 0, 2.5, 5, 7.5, and 10% of relative humidity, respectively. While, in the case of CO, the responses were 92, 61, 50, 41, and 29% for 0, 2.5, 5, 7.5, and 10% of relative humidity, respectively. Even though both $NO_2$ and CO responses decrease with the increase of relative humidity, the CO responses decrease more steeply (from 92 to 29%) than the $NO_2$ response (from 94 to 70%). On the other hand, the selectivity ratio between $NO_2$ and CO (i.e., $S_{NO2}/S_{CO}$), displayed in Figure 7b, indicates that this ratio, being in the favor of $NO_2$, increases from 1.02 to 2.41 when the relative humidity increases from 0 to 10%.

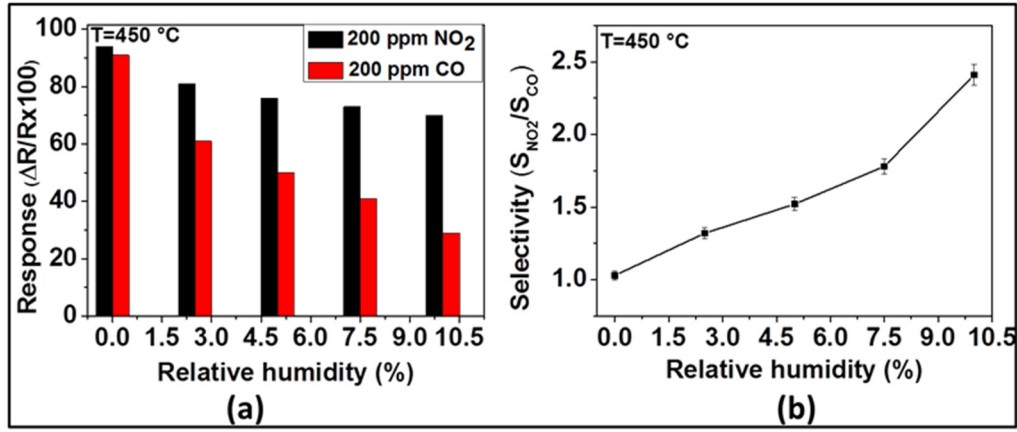

**Figure 7.** Response of ZnO thin film to 200 ppm of $NO_2$ and CO (**a**) and its selectivity ($S_{NO2}/S_{CO}$) (**b**) at 450 °C in the presence of different relative humidity.

In order to confirm the influence of humidity on sensor selectivity toward $NO_2$, we have performed two additional experiments. The first experiment was performed in dry air and the second in 10% RH. We measured sensor response successively to 100 ppm of $NO_2$ only, then to 100 ppm $NO_2$ and 100 ppm CO together, and finally to 100 ppm of CO only. As observed in Figure 8, when only $NO_2$ is introduced, the resistance increases (e.g., as an oxidizing gas, $NO_2$ is introduced, a reduction reaction occurs by taking electrons from the surface of material). When only CO is introduced, the resistance decreases (e.g., as reducing gas, CO is introduced, an oxidation reaction takes place by giving electrons back onto the material surface). When both CO and $NO_2$ gas are introduced, we observed that the resistance increases like in the presence of $NO_2$ even though its absolute value is lower than in the presence of $NO_2$ only. Due to the opposite behavior observed when only $NO_2$ and only CO are introduced, the neutralization should be expected if there was no selectivity toward a specific gas. The change of the resistance in the direction of $NO_2$ indicates the selectivity toward that gas. In the presence of humidity (Figure 8a), the increase of resistance after introducing CO and $NO_2$ together is higher than what is observed in the absence of humidity (Figure 8b), which means that the sensor is more selective to $NO_2$ in the presence of humidity.

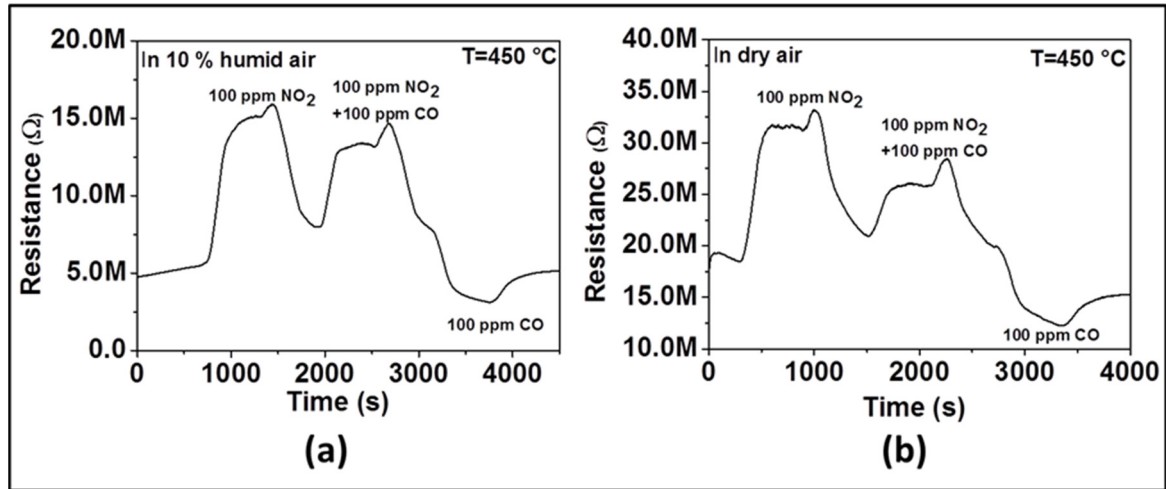

**Figure 8.** Dynamic response of ZnO thin film to 100 ppm $NO_2$, 100 ppm $NO_2$ + 100 ppm CO, and 100 ppm CO at 450 °C in humid air (**a**) and dry air (**b**).

The decrease of $NO_2$ and CO sensor response in the presence of humidity can be explained as follows: in the absence of humidity, only adsorbed oxygen occupies all available active sites on the

surface of the sensing layer, whereas in the presence of humidity, some active sites will be occupied by water molecules. In other words, in dry condition, more surface adsorbed oxygen implies that more target gas will react with the sensing material, and thus, enhance the sensor response as a result of the reaction between the target gases with the adsorbed oxygen. However, the observed increase of $NO_2$-selectivity can be ascribed to the achievement of higher affinity between –OH species and $NO_2$ than CO. As observed, in the presence of water molecules, $NO_2$ will be readily adsorbed on the surface [25] than CO because of the fact that the Van der Waal forces between O-H and $NO_2$ are stronger than those between O-H and CO [26]. This will lead to the enhancement of $NO_2$-absorption on the sensing layer than that of CO resulting in better selectivity toward $NO_2$. A. Ponzoni et al. [27] reported that in the presence of humidity, $WO_3$ exhibited the similar increase of $NO_2$ response and the decrease of CO response but no explanations have been given to the observed phenomena.

The response and the recovery rates are very important for gas sensors. Accordingly, the sensor response and recovery rates were defined employing the average rate change of sensor response during the response and recovery processes. For that, the calculation is carried out by dividing 90% of the full sensor response with the respective response and recovery times ($0.9S/t_{res}$ and $0.9S/t_{rec}$).

Regarding the above observations and consideration of influence of OH species on adsorption kinetics, we have investigated the effect of humidity on response and recovery rates of our sensor. In order to do so, firstly the sensor response is measured and the response and the recovery times (the response time $t_{res}$ and recovery time $t_{rec}$) were calculated by taking 90% of the response achieved during response and recovery of the sensor toward 200 ppm $NO_2$ at 450 °C under application of different relative humidity contents. As Figure 9 displays, the response and recovery rates decrease as the relative humidity increases. That means, in the presence of humidity, the sensor is taking more time to reach the relevant resistance value on inducing $NO_2$ and back to the initial baseline resistance on ceasing the $NO_2$ flow to recover. This delay can be explained by the fact that the water molecules induced through the presence of humidity are adsorbed on the surface and thus reduce the access of $NO_2$ gas to the active reaction sites available on the surface of the sensing material, in this case, ZnO. Due to these controversial reactions, the response rate may decrease considerably. Owing to the fact that there is a strong interaction between O-H and $NO_2$, desorption of $NO_2$ becomes retarded on ceasing the gas flow which leads to the reduction of the recovery rate (e.g., increase of the recovery time).

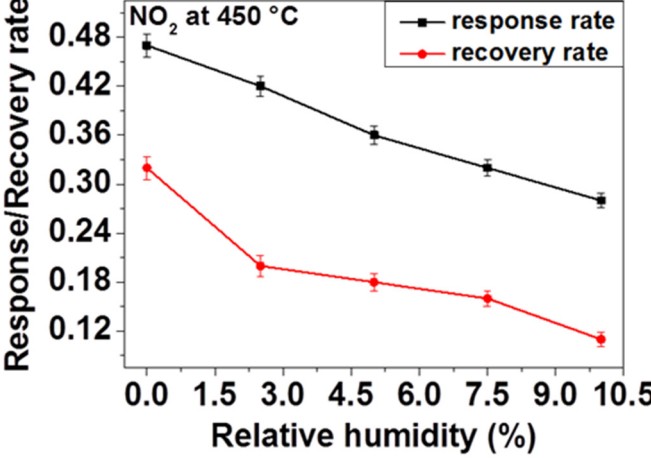

**Figure 9.** $NO_2$ response and recovery rate of ZnO thin film at 450 °C in the presence of different relative humidity.

*Sensing Mechanism*

The sensing mechanism is based on the change in the resistance values achieved on interaction of the target gas with the sensing material by the adsorption and desorption processes of oxygen

molecules at the surface of sensing oxides [28]. It is well known that the reaction between metal oxide and adsorbed gas is a dynamic and reversible process and both kinetics and equilibrium depend on the temperature. In general, when an n-type metal oxide semiconductor gas sensor is exposed to air, oxygen molecules will be adsorbed onto the surface of sensing material and ionize into species such as $O_2^-$, $O^-$, and $O^{2-}$ by reducing electrons at the valence band of the n-type semiconductor and this will form a depletion layer on the surface [29]. In this process, oxygen molecules act as electron acceptor to decrease the electron concentration and, thus, increase the resistance of the sensor. ZnO is n-type semiconductor with superficial electron carriers. Upon $NO_2$ exposure (as an example to oxidizing gas), at an appropriate temperature, these gas molecules will react with the adsorbed oxygen ions on the surface of sensing material and take electrons away, thus, increasing the resistance of the n-type ZnO sensor. The reactions involved are shown in Equations (1) and (2). When the $NO_2$ concentration increases, the number of electrons extracted from the conduction band of ZnO increases resulting in the increase of the electrical resistance and the sensitivity. In the presence of NO, the mechanism is almost the same as $NO_2$. The reaction involved is shown in Equation (3). Upon CO exposure (as an example to reducing gas), the oxidation reaction will take place with adsorbed oxygen ions on the surface and the electrons will be released, thus decreasing the resistance of the n-type ZnO sensor. The reaction involved is shown in Equation (4).

$$NO_2 \ (g) + e^- \rightarrow NO_2^- \tag{1}$$

$$NO_2 \ (g) + O^-(ads) + e^- \rightarrow NO_2^- + O^- \tag{2}$$

$$NO(g) + e^- \rightarrow NO^- \tag{3}$$

$$CO + O^-(ads) \rightarrow CO_2 + e^- \tag{4}$$

## 4. Conclusions

ZnO thin film was successfully deposited by a thermal spray-CVD technique using zinc acetate/methanol solution as a precursor. The sensing properties of the sensor having this ZnO thin film as sensing material were tested between 400 and 500°C toward various gases (e.g., $NO_2$, CO, and NO). According to the XRD and SEM results, single phase of hexagonal wurtzite structured ZnO with hexagonal flake-like morphology was obtained. At 450 °C and in dry air, ZnO thin film exhibited good sensor properties with respect to $NO_2$ and CO. The humidity was found to have a great influence on sensor performance. As the amount of humidity increases from 2.5 to 10% RH, the selectivity ratio of ZnO thin film to $NO_2$ against CO increases from 1 to 2.4. This can be explained by the presence of stronger Van der Waal interactions between OH species and $NO_2$ than those between OH and CO. This technique opens up a new opportunity to synthesize ZnO for $NO_2$ selective gas sensors and enables its use at high temperatures by benefiting the good interaction between OH and $NO_2$.

**Author Contributions:** R.L.F.: Conceptualization, data curation, formal analysis concerning XRD, SEM analysis, methodology and investigation concerning sensor testing, writing–original draft preparation; B.S.: supervision, project administration, funding acquisition, validation, visualization, writing—review and editing.

**Funding:** This research grant that was funded toward R. Lontio Fomekong in the frame of the DLR-DAAD Fellowship program is acknowledged.

**Acknowledgments:** The authors thank Alexander Francke of the central analytic group (ZAN) at the Institute of Materials Research of DLR for XRD measurements.

**Conflicts of Interest:** The authors declare no conflict of interest.

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
