# Peer review of "Influence of Humidity on NO2-Sensing and Selectivity of Spray-CVD Grown ZnO Thin Film above 400 °C"

_chemosensors, doi:10.3390/chemosensors7030042_

Round 1

Reviewer 1 Report

This manuscript describes work that is not novel and is very routine. The effect of humidity on the sensitivity of ZnO-based gas sensors toward NO2 is well known - see Umar et al., Nano-Micro Lett., (2015), 7, 97-120. The aim or reason of the work is also unclear and the experimental conditions investigated are not sufficient for development of a sensor for exhaust gas sensing in an exhaust. Therefore, I recommend reject.

Author Response

Please find our reply in the attached file.

Reviewer 2 Report

I liked the idea of the authors to compare the sensitivity of a semiconductor sensor with to different gases depending on the air humidity. However, before publication, the article requires clarification of some technical issues and strengthening of the discussion.

1. The decomposition reaction of zinc acetate must be rewritten correctly.

2. For thin films, their thickness is a very important characteristic. It is necessary to provide information about the thickness of the films and their uniformity over the substrate surface. 

3. Information should be provided on how gas mixtures with a given concentration of target gases and humidity were created.

4. The authors' statement that "the weakness at the diffraction intensity suggests that the particle size is small or the crystallinity is poor" is not correct. Such a quality of the diffractogram may be due to the low film thickness. I suggest that the authors repeat the diffraction experiment with a slower rate in order to lower the signal-to-noise ratio.

5. Black scale bars on Figure 2 are almost invisible. 

6. It is necessary to show the dynamic response of ZnO thin films to CO and NO gases.

7. In the caption of  Fifure 5 it is written that the measurement temperature is 200 C. is it correct?

8. The main point. To confirm the authors' conclusions about the increase in selectivity with changes in humidity, it is necessary to make experiments with gas mixtures containing NO2 and CO in various concentrations (for example, by varying the concentration of CO at a constant concentration of NO2 and vice versa).

9.The discussion of the sensor mechanism is very brief. The formation of a sensor response when interacting with CO and NO is not discussed at all.

10. Discussion of the reasons for the increase in selectivity with increasing humidity requires additional experiments (for example, operando DRIFT) or the use of literature data.

Author Response

(The authors gave the same response as above.)

Reviewer 3 Report

General Comment: The manuscript entitled “Influence of humidity on NO2 sensing performance of ZnO thin film grown by thermal spray-CVD”. The manuscript is well written and results are interesting, however, Overall, this manuscript has many flaws and it requires revision before publication. The manuscript can and should be further improved by taking the following aspects into consideration.

1.     An introduction is poorly written, please include the scope and limitations of the ZnO based sensors, their advantages, and disadvantages since ZnO is very common material and it is hard to find the novelty in the paper.

2.     Abstract: Please include some important parameters such as response and recovery time, the selectivity of the sensor compared to other oxidizing and reducing gases.

3.     Authors mentioned that “To our best knowledge for the first time, we report in this work, the influence of relative humidity on sensing performance of ZnO thin film beyond 400 °C.” Here the question is what is the role of humidity at 400C, Humidity generally affects the sensor performance at low temperature or room temperature. Please clarify this statement.

4.     How gas concentration has been calibrated? Please explain in detail.

5.     About the gas sensor measurement, some parameters are missing like what is the SCCM of the gas? What is the applied voltage?

6.     Figure 1: Please provide (hkl) and JCPDS number in the XRD plot.

7.     It is better to mention in the introduction that how the selectivity could be enhanced? include some strategies to improve selectivity and cite relevant references: Some suggested review article to cite in the intro. Part: Adv. Mater. 2016, 28, 795–831, Microchimica Acta, 185(2018) 213, Adv. Funct. Mater. 2017, 27, 1702168.

8.     How the prepared sensor is better than room temperature sensors. Please take a look and justify this point by citing these articles: Trans. Nonferrous Met. Soc. China 29(2019) 143−156, J Sol-Gel Sci Technol (2018) 88: 322, Microchim Acta (2019) 186: 418, Nano-Micro Lett. (2015) 7: 97., Sensors & Actuators: B. Chemical 287 (2019) 191–198.

9.     The quality of some figures is very poor and needs to be enhanced.

10.  It is better to check and correct the font size of x and y-axis of all figures in the manuscript. It should be the same.

11.  Long-term stability of the sensor is missing.

12.  Please include the response curve to understand the selectivity towards different gases.

13.  Reproducibility of the sensor is very important, please include at least 5 cycles for a particular concentration.

14.  Please provide response and recovery time Vs gas concentration with an error bar.

Author Response

(The authors gave the same response as above.)

Round 2

Reviewer 1 Report

Title: Influence of humidity on NO2-sensing and selectivity of spray-CVD grown ZnO thin film above 400 °C

Saruhan et al.

The authors have made sufficient changes to the manuscript, based on the reviewers' comments, for me to decide that the manuscritp should be accepted after minor revisions. My minor revision suggestions are as follows:

1) Page 1, line 10 and page 2, line 58: In both instances "...high surface to volume ratio..." should be replaced with "...high surface area to volume ratio..."

2) Typos:

Page 2, line 65 and Page 10, line 303: In both instances "synthetize" should be corrected to "synthesize". Page 9, line 245: "Van der Wall" should be corrected to "Van der Waals".

3) The authors should add a brief comment on how the alumina substrate decorated with interdigitated electrodes was fabricated.

Author Response

Page 1, line 10 and page 2, line 58: In both instances "...high surface to volume ratio..." should be replaced with "...high surface area to volume ratio..."

Answer: Thanks for the comment. These lines are corrected in the revised manuscript.

2) Typos:

Page 2, line 65 and Page 10, line 303: In both instances "synthetize" should be corrected to "synthesize". Page 9, line 245: "Van der Wall" should be corrected to "Van der Waals".

Answer: Thanks for the comment. These typos are corrected in the revised manuscript.

The authors should add a brief comment on how the alumina substrate decorated with interdigitated electrodes was fabricated.

Answer: Thanks for the comment. The authors have added a brief comment in the revised manuscript (pls see between the lines 98 and 102 on page 3) how the substrate decorated with IDE was fabricated.

Reviewer 2 Report

The authors basically made the necessary changes to the article in accordance with my comments. In my opinion, the article may be accepted for publication.

Author Response

The authors thank the reviewer for appreciation.

Reviewer 3 Report

All the comments addressed properly, the manuscript is ready for the publication.

Author Response

The authors thank the reviewer for the appreciation.